# Systematic review of post-COVID condition in Nordic population-based registry studies

Jan Peter William Himmels [1] ✉, Karin Magnusson[1,2] & Kjetil Gundro Brurberg [1,3]

The long-term effects of COVID-19, known as post-COVID condition (PCC), are still not fully understood. This systematic review synthesizes findings from Nordic registry studies to highlight long-term outcomes after COVID-19 infection. Twenty-two studies, primarily reflecting the pre-omicron and early vaccination phases, reveal increased primary care use for respiratory issues and fatigue in the sub-acute and chronic phases, with PCC incidence estimated below 2% in the general population. Most individuals returned to work within three months post-infection, and the risk of new neurological or mental disorders did not exceed that in patients with other infections. The review demonstrates the value of high-quality Nordic health registries in capturing reliable, population-wide data, though generalizability may be limited to similar healthcare systems. Findings suggest the need for targeted follow-up in patients with severe COVID-19, particularly those requiring intensive care, to manage potential new-onset diseases and guide resource allocation in the pandemic's endemic phase.

The COVID-19 pandemic has had a profound impact on global health systems, economies, and societies[1]. Since its emergence, millions of cases have been reported worldwide, with substantial morbidity and mortality[2]. Acute COVID-19 symptoms and outcomes have been extensively studied, but understanding the long-term effects of the disease, often referred to as "long COVID" or post-COVID condition (PCC), has been more challenging. It is defined as the continuation or development of new symptoms three months after the initial SARS-CoV-2 infection, with these symptoms lasting for at least two months with no other explanation[3].

Despite being extensively studied in survey data and clinical cohort or case-control studies, data about the long-term consequences of COVID-19 remains inconsistent and inconclusive. Estimates for prevalence of post covid condition (PCC) in the UK are around 3.1%, with WHO Europe reporting the range 10 and 20%[3,4]. Inconsistencies in prevalence arise due to differences in case definitions and diagnostic strategies, but also as a result of study design. Observational studies that recruit participants based on their willingness to respond may be exposed to selection bias and recall bias when reporting typical post-covid symptoms[5–7]. Further, many cohort studies of selected patient populations lack day-to-day data, and hence, cannot account for important time-varying confounding, like variations in healthcare use over seasons.

In register-based observational research applied to all individuals within a geographical unit, all these three types of biases are reduced. Misclassification bias is minimised by the greater reliance on objectively measured data, typically requiring clinical examination and doctor judgement for reporting post-acute COVID complaints. Selection bias is reduced by the inclusion of everyone, not only those willing to participate. Confounding bias can be reduced through thorough adjustment for objectively measured socioeconomic factors and calendar time, which is often addressed through panel data setups in registry studies.

Nordic countries, known for their comparable, comprehensive healthcare systems and high-quality population-based registries, are uniquely positioned to quantify burden and contribute valuable

[1]Norwegian Institute of Public Health, Oslo, Norway. [2]Lund University, Faculty of Medicine, Department of Clinical Sciences Lund, Orthopaedics, Clinical Epidemiology Unit, Lund, Sweden. [3]Department of Life Sciences and Health, Oslo Metropolitan University, Faculty of Health Sciences, Oslo, Norway. ✉e-mail: jahi@fhi.no

insights into the long-term effects of COVID-19[8,9]. Registry-based studies from Nordic countries have already provided critical data on various aspects of the COVID-19 pandemic, including infection rates, vaccination effectiveness, and mild and severe short-term outcomes[10–12]. These data were crucial for the management of the COVID-19 pandemic in its acute phase. However, existing data from these registries have never been combined to provide knowledge about the pandemic beyond three months.

In this systematic review, we aim to compile and analyse data from Nordic registry studies to provide a comprehensive overview of the long-term effects of COVID-19. By synthesizing evidence from the Nordic countries, this review seeks to identify the incidence of PCC, the associated healthcare use and sick leave, as well as new onset diseases. The findings can inform healthcare providers, researchers, and policymakers about the enduring impact of COVID-19 and guide future research and healthcare strategies to mitigate its long-term effects.

## Results

### Results of the literature search

Searches of two databases identified 2933 publications. Of 1803 unique abstracts screened, 45 were reviewed in full text. JH and KGB independently screened all titles and abstracts in EPPI reviewer and Covidence[13,14], and 22 unique studies matched our inclusion criteria, in the PRISMA flow diagram[15] (Fig. 1). Studies excluded in the full text review and reasons for exclusion are shown in Supplementary Table 1.

Table 1 provides an overview of all included studies, and main study characteristics. Out of the 22 studies that met our inclusion criteria, eight originated from Denmark, nine from Norway and five from Sweden. Study size varied from 7640 to 4,888,615 participants. While most studies primarily involved adults, six included children or adolescents. The duration of follow-up across all included studies varied between three months and two years. Five studies documented healthcare use. Five studies examined return to work or length of sick leave, while ten studies reported incidence of PCC or new onset diseases. All studies sampled patients during 2020 and 2021, four studies also sampled patients from 2022 (Fig. 2). COVID-19 vaccines became wider available from 2021, reaching near universal coverage in the Nordics by the end of 2021. Hence, the participants in the included studies mostly represent those infected with earlier, more clinically harmful variants, and patients who were not fully vaccinated, both contributing to more pronounced long-term effects[16,17]. An overview of how authors defined their outcomes, populations and controls can be found in the Supplementary Table 2.

### Quality assessment

We evaluated studies based on adherence to rigorous research principles for registry studies, focusing on key domains including registry validity, epidemiological bias, data quality, participant selection, outcome measures, confounding variables, statistical analysis, and reporting transparency. Potential concerns and consequences of various methodological approaches were narratively discussed in the description of results. Our evaluation of the studies did not influence the final selection of studies included in our review.

### Healthcare utilisation and reason for healthcare use

Eight studies reported data on healthcare (HC) use following COVID-19, six studies from Norway and two from Denmark[18–25]. Both hospitalised or non-hospitalised individuals with a positive Polymerase Chain Reaction (PCR) for SARS-CoV-2 were included, with study sizes ranging between $N = 87,288$ and $N = 2,348,831$. PCR-positive cases were compared with negative controls (negative PCR in one control group and non-tested in another control group). Individuals in these eight studies were included from February 2020 until December 2021. All studies except one analysed the pre-omicron and pre-vaccination phase of the pandemic. The Norwegian studies reported all-cause or cause-specific primary or specialist care visits, or complaints recorded by the general practitioner. The Danish studies considered all-cause

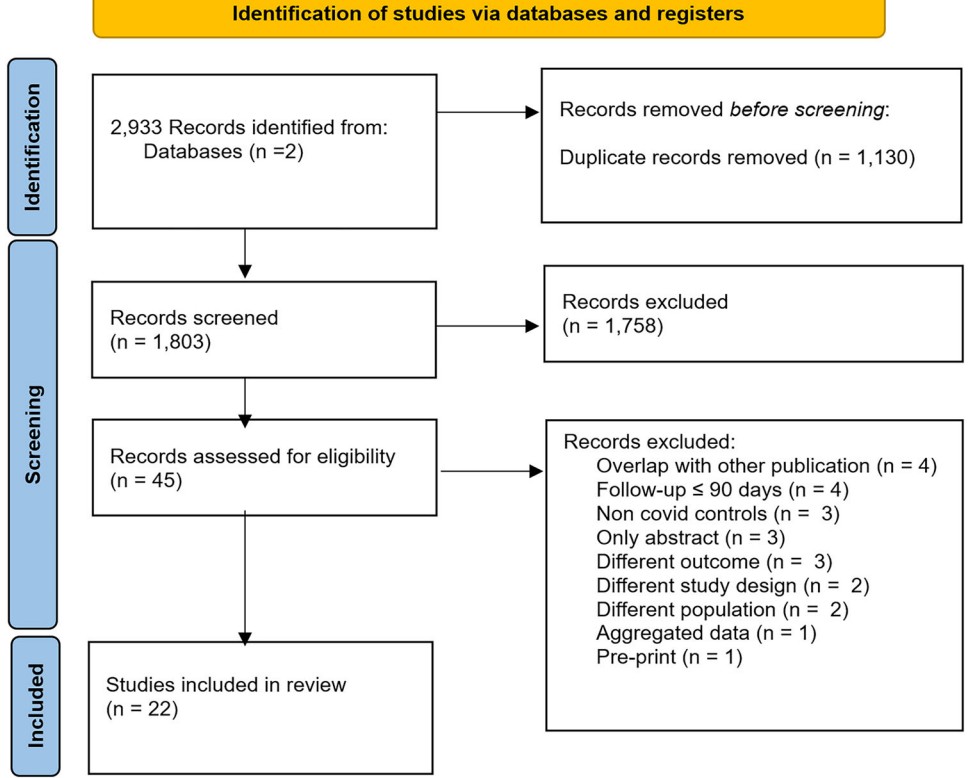

**Fig. 1 | PRISMA flow diagram.** Flow diagram summarises the number of studies excluded at each stage.

**Table 1 | Major characteristics of studies included in the systematic review**

| Author | Country | Outcome[a] | Population type | Participants (n) |
|---|---|---|---|---|
| Abzhandadze[27] | Sweden | SL | People with sickness benefits for COVID-19 | 11,902 |
| Andersson[37] | Denmark | NOD | Adults ≥ 50 years | 2,430,694 |
| Bygdell[36] | Sweden | PCC | Adults with previous COVID-19 | 506,107 |
| Bygdell[35] | Sweden | PCC | Children with previous COVID-19 | 162,383 |
| Gronkjaer[38] | Denmark | NOD | Children and adults | 4,888,615 |
| Hedberg[17] | Sweden | PCC | Children & adults ≥ 1 year | 473,359 |
| Hetlevik[22] | Norway | PCS | Adults | 539,603 |
| Jacobsen[11] | Denmark | SL | Working-age population | 7640 |
| Kildegaard[21] | Denmark | HCU, PCC | Children | 995,504 |
| Lund[20] | Denmark | HCU, NOD | Children and adults with previous COVID-19 or negative test | 91,187 |
| Magnusson[23] | Norway | HCU, PCS | Children | 706,885 |
| Magnusson[24] | Norway | PCS | Adults | 1,323,145 |
| Magnusson[25] | Norway | PCS | Adults | 2,348,831 |
| Mkoma[33] | Denmark | PCC | Adults with previous COVID-19 | 2,287,175 |
| Nersesjan[40] | Denmark | NOD | Adults with previous COVID-19, negative and untested controls | 4,152,792 |
| Reme[34] | Norway | PCC | Adults (30 to 70 years) with previous COVID-19 | 214,667 |
| Rømer[39] | Denmark | NOD | Adults with previous COVID-19 and matched negative controls | 2,039,204 |
| Skei[28] | Norway | SL | Working-age population admitted to hospital with sepsis | 35,839 |
| Skyrud[18] | Norway | HCU | Adults | 1,401,922 |
| Skyrud[26] | Norway | SL | Working-age population with work contract | 1,177,274 |
| Skyrud[19] | Norway | HCU | Adults hospitalized for COVID-19 or other respiratory tract infections | 87,288 |
| Spetz[29] | Sweden | SL | Working age population | 37,420 |

[a]*HCU* Healthcare use, *SL* Sick leave/Return to work, *PCC* Post-COVID condition, *PCS* post-COVID symptoms, *NOD* new onset disease.
The terms and definitions used by authors for the outcomes are provided in detail in Supplementary Table 2.

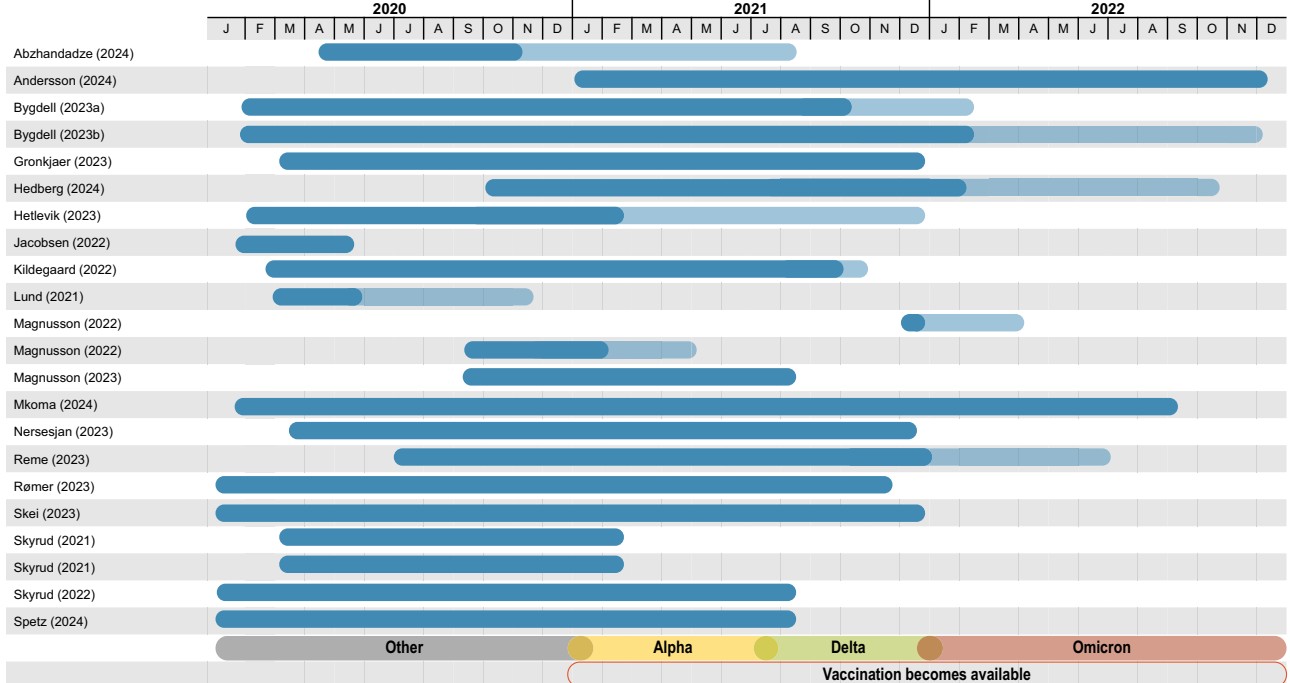

**Fig. 2 | Overview of study timeframes.** Overview of timeframes, times from patient sampling (dark blue) and extended follow-up (light blue) beyond sampling period. Vaccination, virus variants dominating at the respective timeframes are colour coded.

primary and specialist care use as well as drug prescription and specific hospital-based diagnoses. Seven of the eight studies focused on non-hospitalized cases, i.e., mild COVID-19 cases.

Five studies reported primary and specialist HC utilisation, three in adults[19,20,26] and two in children[21,23]. Adults with mild COVID-19 (non-hospitalized in acute phase) in Denmark and Norway had more general practitioner or primary care visits up to six months after testing positive than controls[18,20]. Similar results were reported for children[21,23]. Norwegian studies reported that the group difference decreased over the first six months after a positive test, whereas the

Danish studies did not analyse the development over time. Adult COVID-19 patients requiring ventilation support at the hospital were reported to have higher use of physiotherapy for six to nine months when compared to influenza patients with similar severity level[19].

Respiratory complaints and diagnoses were the main cause for contact with primary or specialist healthcare following mild SARS-CoV-2 infection. Lund et al. demonstrated increased use of bronchodilating agents and more frequent hospital diagnoses of dyspnoea in SARS-CoV-2 positive adults compared to controls[20]. Skyrud et al. also reported that respiratory complaints were the main reason for the increased frequency of primary care visits that was seen in COVID-19 patients compared to negative controls[18].

Three studies analysed symptoms as recorded by a general practitioner in primary care[22,24,25]. Respiratory complaints, general complaints (e.g., fatigue) and neurological complaints (e.g., memory loss or concentration problems) were more frequently reported after testing positive than after testing negative or not being tested. The differences between the groups were largest in the acute and sub-acute phase 0 to 4 months and decreased into the chronic phase. In analyses of reported symptoms over time, neurological problems seemed less common among SARS-CoV-2 positives compared to negatives (HR ~ 1.00–1.40) than respiratory complaints and fatigue (HR ~ 1.20–2.80)[22,24].

One study compared individuals with the omicron variant with individuals with the delta variant, in addition to the comparison with negative controls[24]. Findings implied that omicron and delta had a relatively similar increase in doctor-reported symptoms according to diagnostic codes in primary care for the 14th to 126th day after positive test, when compared to the same period after negative test[24].

In summary, Nordic registry studies consistently report that primary care use is increased after mild SARS-CoV-2 infection for children and adults, particularly in the sub-acute phase up to three months, but also in the chronic phase from three months and beyond. The increase is mainly due to respiratory complaints and fatigue.

## Return to work

Five included studies provided data on return to work[11,26–29], while three were excluded due to substantial overlap of analysed populations[30–32]. Still, however, some overlap persists due to similar study timeframes within included studies from the same country. Two studies each covered the Swedish and Norwegian population, and one the Danish. Working age participants were included between 1 January 2020 until 31 August 2021, reflecting the pre-omicron and early vaccination period. Mainly involving PCR-tested participants, the study sizes varied from N = 7466 to N = 1,177,274 with follow-up ranging from three months to two years. The studies included all COVID-19 positive participants within their respective countries, except for one focusing solely on intensive care units (ICU) patients[28] and another on those receiving a sick leave[27]. The Swedish and Danish studies analysed sick leave duration, and variables affecting the return to work. The Norwegian studies compared sick leave patterns by admission status, and with non-COVID sepsis patients.

In Sweden, 5.7% of the COVID-positive population took sick leave after their first positive COVID-19 test, with a median duration of 31 days[29], and 35 days[27]. Return to work at ≥12 weeks was 99.4% for men and 99.2% for women. Of those initially on sick leave, 3% were still on sick leave at the one year follow-up[27]. Among working age Danes testing positive by May 2020, 81.9% returned to work within four weeks, with 98.4% back at work by six months[11]. In Norway, a substantially elevated sick leave was observed 1–4 weeks after a positive test. The elevation gradually returned to baseline, although not fully for women aged 45–70[26]. For severely ill patients requiring ICU care, 66.9% returned to work within six months, and 77.8% within a year after hospital discharge[28].

All studies analysing sick leave following a COVID-19 infection investigated factors influencing sick leave duration. Most consistently, older age, being female[11,26–29], and initial disease severity[11,26–28] correlated with extended duration before return to work. One study found several demographic and socioeconomic variables of relevance, although less pronounced or reversed among hospitalised patients[27].

Sick leave after COVID-19 was higher than seen in influenza[11]. The probability of sustainable return to work was lower in ICU patients compared to non-ICU patients (HR 0.56; 95% CI 0.52–0.61) and higher in patients with COVID-19-related sepsis than in other sepsis patients (HR 1.31; 95% CI 1.15–1.49)[28].

Overall, most COVID-19 patients return to work by three months, whereas a small proportion, typically less than 2%, remain on sick leave after six months. Older age, female sex and severe initial illness correlate with longer sick leaves.

## Post COVID condition

Six studies from Denmark[21,33], Norway[34] and Sweden[17,35,36] reported incidence of PCC. The Danish and Norwegian studies were nationwide, whereas the Swedish studies were regional covering Stockholm[35,36] and Västra Götaland[35,36]. Despite overlapping patient samples in studies from the same country, differences in methodology offered complementary results. Reported incidences ranged from 0.12%[21] to 2.0%[36] with lower-end rates among children[17,21,35]. In adults, stricter case definitions for PCC (e.g., symptom duration ≥ 3 months) and longer data collection period (e.g., inclusion of omicron cases) resulted in incidence estimates between 0.20%[33] and 0.63%[17]. The 2.0% estimate was attributed to a study applying looser case definition (PCC could be set ≥ 28 days after positive test) primarily involving patients infected with the alpha and delta variants[36]. The incidence of PCC was markedly higher among patients who had been hospitalised with COVID-19 than among non-hospitalised ones[17,33].

## New-onset disease

Five Danish studies investigated the risk of new-onset diseases following COVID-19, but differed regarding study period, length of follow-up and scope. An early phase study found no evidence linking SARS-CoV-2 infection to the onset of chronic diseases such as pulmonary diseases, diabetes mellitus or neuropathies[20]. Subsequently, more extensive studies have investigated the association between SARS-CoV-2 and infectious diseases[37], neurological disorders[38] and mental disorders[39,40].

Andersson and coworkers compared post-acute risk of non-COVID-19 infection hospitalisations in adult patients testing positive (n = 930,071) and negative (n = 1,500,623) for SARS-CoV-2[37]. Calculated incidence rate ratios for various groups of infections found no evidence for increased non-COVID-19 hospitalisation rates in the SARS-CoV-2 positive group, neither during the post-acute (29–180 days) nor the long-term (>180 days) follow-up.

In a study by Grønkjær and coworkers[38], the risk of any new neurological disorders was compared between patients who tested positive for SARS-CoV-2 from March 2020 to December 2021 (n = 675,961), those who tested negative (n = 3,655,688), and those who were not tested (n = 556,966). SARS-CoV-2 positive tests were associated with more new-onset neurological disorders when compared to SARS-CoV-2 negatives (HR 1.11; 95% CI 1.07–1.16), but this association vanished when positives were compared to patients treated for other lung infections (HR 0.84; 95 % CI 0.80–0.89). Analyses restricted to patients hospitalised with COVID-19 revealed a significant connection between COVID-19 and new-onset neurological disorders compared to the general population (HR 2.92; 2.64–3.23), but not when compared to patients hospitalised with other lung infections (HR 1.06; 0.94–1.20)[38].

Two studies explored a potential link between COVID-19 and long-term mental problems[39,40]. Data collection strategy and period were quite similar in the two studies, and patients were followed for up to 1.8 years after their first positive SARS-CoV-2-test. Both studies

demonstrated an increased use of anxiolytics among patients who had tested positive for SARS-CoV-2 as compared to negative controls. However, this increase was not followed by more psychiatric admissions[39] or more new-onset mental disorders[40] in analyses embracing all SARS-CoV-2 positive patients. Stratified analyses suggested a possible non-linear age effect[39,40]. Furthermore, patient hospitalised with COVID-19 had significantly higher risk of new-onset mental disorders when compared to the general population (HR 2.49; 95 % CI 2.07–3.00), but comparable risk as patients who were hospitalised with other respiratory infections (HR 1.03; 0.82–1.29)[40].

## Discussion

Nordic registry studies report an incidence of PCC in the general population to be less than 2%. This corresponds well with the finding that most COVID-19 patients return to work three months post-infection, with less than 2% of all COVID-19 patients still on sick leave after six months. The reviewed studies also consistently report increased primary care use following mild SARS-CoV-2 infection in both children and adults, particularly during the sub-acute phase (28 days to 3 months) and continuing into the chronic phase (beyond 3 months). This increase is primarily attributed to respiratory complaints and fatigue. Older age, female sex, severe initial illness, and early virus variants are associated with longer sick leaves and higher incidence of PCC. The risk of new-onset neurological or mental disorders following COVID-19 was generally not higher than for patients experiencing other infections.

### Comparison to previous studies

Our systematic review represents the first and most comprehensive overview of long-term effects following COVID-19 that is solely based on registry-based data. We report a prevalence of PCC of less than 2%, while in previous studies, prevalence estimates range from 3.1% up to 20%[3,4]. There are important differences in the interpretation of these estimates. Where estimates of prevalence of PCC in previous studies are typically based on self-report, which induces bias due to self-selection and recall[5], the current study covers entire populations in entire geographical units with PCR testing available for everyone and routinely registered medical records for everyone. Some studies also included temporary workers or migrants, but there might be variations across studies, time periods and across the different Nordic countries in the extent to which these were included. Observational studies based on survey often include a non-random sample of the target population, potentially resulting in misleading associations[6]. Thus, selection biases might have affected conclusions in previous systematic reviews, but also in official statistics[3–5]. In our study, differences in diagnostic coding practices of PCC across the Nordic countries might have affected the results. However, studies comparing complaints included in diagnostic codes available prior to the pandemic (e.g., ICPC-2 code A04 fatigue), also found a 2% increase in individuals testing positive for SARS-CoV-2[25]. These data support the interpretation of a PCC prevalence of up to 2%, independent of diagnostic coding practices.

The register-based studies that are reviewed in this study typically include rigorous sensitivity analyses to explore the robustness of their results to different selection mechanisms. For example, there might be selection mechanisms in PCR testing patterns, where particularly health-concerned individuals are testing themselves more often than less health-concerned individuals. To face such challenges, register-based studies allow for the inclusion of a randomly selected negative and untested control subjects. The latter are typically assigned a random test date[6,24], keeping in mind that there might be both false positives and false negatives in this population.

A final and important difference between survey-based vs register-based studies is the ability to adjust for time-variant confounding. In survey data, patient responses are typically collected at predefined time points or when it is convenient for both the researcher and the patient to respond. In contrast, in routinely collected PCR-testing and healthcare data, a lack of record on a certain date is equal to not having the event under study at that certain date. For example, there is typically less transmission and less healthcare use during weekends and holidays, which can easily be controlled for in statistical models applied to panel data. Thus, the use of complete coverage register data in research enables a thorough adjustment for seasonal variations in virus transmission and healthcare use.

### Estimates of societal burden

Because of the above-described strengths, the current study has significant implications for understanding the prevalence of post-COVID-19 condition and its burden on healthcare systems. As of 2023, the combined population of the Nordic countries (Denmark, Finland, Iceland, Norway, and Sweden) is approximately 27 million. Notably, there were only minor differences between countries and methodologies, allowing for a reliable quantification of the chronic COVID-19 burden on healthcare systems and the workforce. Assuming that 90% ($N = 24.3$ million) of the Nordic population has been infected with SARS-CoV-2, it can be projected that 2% ($N = 486,000$) may have experienced or are still experiencing long-lasting complaints consistent with PCC. A similar number of individuals may still be on sick leave six months after the initial infection.

It is crucial to note that the projection above is based on studies conducted during the earliest phases of the pandemic. During this time, testing was widely available and free, no vaccines were yet developed, and the predominant virus variants caused relatively severe acute illness in many individuals. In later phases, testing became less accessible and is now neither recommended nor mandated. Simultaneously, vaccination rates have increased, and newer virus variants tended to cause milder disease[41,42] with studies suggesting a lower risk of developing PCC[17]. Hence, the prevalence estimates and the projected societal burden reported in this review may be overestimated, but there are also reports indicating that later virus variants result in an equal increase in complaints as earlier variants[24]. As such, the estimates presented in this review might still be valid as of today.

Finally, it is important to recognize that medical records of complaints and diagnoses may not fully capture how a patient feels. The individual burden of PCC cannot be accurately estimated without closely monitoring symptom changes on a day-to-day basis. For example, patients may stop visiting their general practitioner when no treatment is provided, resulting in an underestimation of the true individual burden of PCC. Still, the included register studies did capture a group of individuals with long-lasting complaints. Many of these patients needs professional care to get back to everyday life. Reducing the reported 2% prevalence to near zero by providing effective clinical care would benefit both individuals and society. Recent publications show promising results implying that health professionals can take important roles in symptom management and treatment of patients with PCC[43–45].

### Strengths and limitations

Our findings are representative of the Nordic countries and other nations with similarly organized health and welfare systems. An important strength of our work is the use of registry data from Nordic countries that represents comparative settings supporting the ability to summarize findings. Further, our search strategy, refined over several years and validated across multiple topics, minimizes the risk of missing relevant Scandinavian studies and strengthens the validity of findings. However, a limitation is that our results may not be generalizable to countries with limited access to testing and healthcare services. Additionally, variations in testing criteria for SARS-CoV-2 over time and between the countries included in our study may have influenced the findings. Despite this, none of the included studies

analysed more than one national population, and any such regional or temporal differences are likely to affect the entire study population uniformly. These differences can be mitigated through appropriate study design and statistical modelling (as described earlier). Furthermore, the outcomes examined can be viewed as proxies for a range of conditions or phenomena, such as healthcare burden, underlying diseases, or reasons for healthcare contact.

In interpreting the findings, it is important to acknowledge that electronic health records in addition to implying type and severity of complaint or disease, also reflects healthcare-seeking behaviour. As such, it is possible that our findings apply mainly to individuals who seek care for their complaints. However, testing for the SARS-CoV-2 virus was free of charge and available to all citizens in the Nordic countries. Thus, healthcare-seeking behaviour might also affect individuals' PCR test patterns, where particularly health-conscious individuals might test themselves more often than non-health-conscious individuals. To take these differences in healthcare seeking behaviour and test patterns into account, some of the studies included in this review studied both a control group where individuals had a negative test and another control group where individuals had not tested themselves and were assigned a random test date. Generally, group differences in post-test healthcare use or diagnoses were greater in analyses including a non-tested control group (potentially overestimated) and smaller when including a group testing negative (potentially underestimated). The over- vs underestimation might be due to the control group's differences in healthcare seeking behaviour relative to the group testing positive, which is likely somewhere in the middle with regard to healthcare seeking behaviour.

In our study, we considered visits to general practitioners or specialists for persistent complaints potentially related to COVID-19 as a proxy for underlying chronic post-COVID-19 conditions. However, we did not conduct specific tests to validate or confirm the reliability of the PCC diagnosis. Given that SARS-CoV-2 is a novel virus, there may be significant variability in diagnostic practices, especially over the study period, as scientific discussions about the criteria for diagnosing PCC were ongoing[46]. Still, the validity of the diagnostic coding in both the Swedish, Danish, and Norwegian patient registers in specialist and/or primary care has generally been reported to be high for a variety of conditions[47–50]. Given the novelty of the virus, it is likely that physicians exercised caution in clinical evaluations and diagnostics. Additionally, medical reporting to national registries is mandated by law in all the countries studied, which significantly reduces the risk of missing data.

Nordic registry studies, characterized by limited selection bias and minimal time-varying confounding, report an incidence of PCC in the general population to be less than 2%. This corresponds well with findings that most COVID-19 patients return to work within three months post-infection, with less than 2% of all COVID-19 patients still on sick leave after six months. While typical post-COVID complaints, healthcare utilisation and sick leave decline rapidly during the sub-acute phase, our results indicate that a significant number of patients continues to experience long-lasting societal burden beyond three months. Our findings may imply that health- and welfare services may need to be up-scaled to address this challenge. Furthermore, improved and earlier clinical management of PCC could help reduce the future strain on healthcare systems and the workforce.

## Methods
When preparing this review, we used the Cochrane Protocol and Review Template for Qualitative Evidence Synthesis[51]. We registered our project protocol with OSF registries 17.06.2024 (https://doi.org/10.17605/OSF.IO/EB3C9).

### Criteria for considering studies for this review
**Types of studies.** We included studies of large national registries, not limited to a single disease, organ or city[9]. We included studies where the recruitment of participant and respective data points were primarily based on registries. Studies where eligibility was based on other selection mechanisms (e.g., questionnaire responders) followed by retrospective collection of data from registries were excluded. We did not exclude studies based on our assessment of methodological limitations, and we did not apply GRADE.

**Topic of interest.** The long-term health effects of COVID-19 in whole populations compared with non-COVID controls as documented in Nordic registry studies (Denmark, Norway, Sweden, Finland, Iceland, the Faroe Island, Åland and Greenland), focusing on the persistence and variety of doctor-reported symptoms, complaints and diagnoses, primary care consultations, specialist care consultations and associated sick leave.

**Inclusion criteria.** Population: Persons tested for COVID-19 (not limited to organ-group, disease, or profession) in the Nordic countries

Exposure: COVID-19

Comparator: Non-COVID-19 controls: different definitions of non-COVID-19 controls were accepted as applicable to the outcome, including COVID-19 test-negative (with or without other infections) or untested

Outcomes: Primary, specialist care consultations, PCC diagnosis, new onset disease, sick leave, return to work

Follow-up period: At least 3 months, however, this requirement did not prevent the included studies from following individuals already from day 0 after PCR testing or being ill with COVID-19.

**Exclusion criteria.** Pre-prints, studies with only self-reported outcomes, studies with less than three months follow-up.

### Search methods for identification of studies
**Electronic searches.** A previously used search strategy developed by an information specialist in consultation with the review authors was adopted to meet a narrow focus on Nordic registry studies only[52].

We searched Epistemonikos (www.epistemonikos.org) for related reviews to identify eligible studies for inclusion, as well as the following electronic databases:

- MEDLINE
- Embase

We did not apply any limits on language or publication date. We searched all databases from 2020 to the latest database update 30.05.2024. See Supplementary Table 3 for the search strategy.

**Grey literature.** We conducted a grey literature search to identify studies or reports not indexed in the databases listed above. We searched the national health, statistics, employment authorities' websites to identify relevant literature. The web searches were expanded as necessary following contact with experts in the field. See Supplementary Table 3 for the search strategy.

**Searching other resources.** We reviewed the reference lists of all the included studies and key references (i.e., relevant systematic reviews). We contacted authors of included studies to clarify published information and to seek unpublished data if necessary. We contacted researchers with expertise relevant to the review topic to request studies that might meet our inclusion criteria. No new studies were identified via our network of researchers within the field.

### Selection of studies
Two review authors [JH, KGB] independently assessed the titles and abstracts of the identified records to evaluate eligibility. We retrieved the full text of all the papers identified as potentially relevant by one or both review authors. Two review authors [JH, KGB] then assessed these

papers independently. We resolved disagreements by discussion or, when required, by involving a third review author. Where appropriate, we contacted the study authors for further information.

We included a table listing studies that we excluded from our review at full text stage and the main reasons for exclusion (see Supplementary Table 1).

## Language translation

All studies were written in English or Scandinavian languages, no translation was needed.

## Data extraction

We extracted information on study country, type and name of registry, participants, follow-up period, primary care consultations, specialist care consultations, sick leave, new onset disease and statistics (e.g., odds ratio, rate ratio, hazard ratio, relative risk, predicted probabilities, margins or differences-in-differences).

## Assessing the methodological limitations of included studies

We evaluated studies based on adherence to sound research principles for registry studies[9], and address flaws where considered relevant in written form. Each study was assessed for key domains, including validity of registry, epidemiological bias, data quality, participant selection, outcome measures, confounding variables, statistical analysis, and reporting transparency. A non-standardised approach without a checklist reflects that we are not aware of a validated and proofed tool for this purpose.

## Data management, analysis and synthesis

We conducted a systematic review, but as the heterogeneous nature of identified study designs and reported outcomes did not allow for meta-analysis, data was synthesised narratively. All figures and tables were created with MS Office.

## Reporting summary

Further information on research design is available in the Nature Portfolio Reporting Summary linked to this article.

# Data availability

All data included in this systematic review are publicly available through the original published studies, which are accessible online. A complete list of sources with links is provided in the references. No additional datasets were generated or analyzed for this review.

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

## Acknowledgements

When preparing this protocol/review, we used the Cochrane Protocol and Review Template for Qualitative Evidence Synthesis (Glenton C, Bohren MA, Downe S, Paulsen EJ, Lewin S, on behalf of Cochrane Person Centred Care, Health Systems and Public Health. Cochrane Qualitative Evidence Synthesis: Protocol and review template. Version 1.4. Cochrane Person Centred Care, Health Systems and Public Health; 2023. Available at: https://zenodo.org/record/5973704).

## Author contributions

Conceptualization (J.H., K.M., K.G.B.), Methodology (J.H., K.G.B.), Writing–original draft (J.H., K.M., K.G.B.), Writing–review & editing (J.H., K.M., K.G.B.).

## Funding

## Competing interests

K.M. declared no financial conflicts of interest. She is a co-author on possibly relevant papers and was excluded from the screening or quality assessment process of these papers. J.H. and K.G.B. declared no financial or other conflicts of interest.
