## [Transparent Peer Review file · Nature Communications]

Systematic review of post-COVID condition in Nordic population-based registry studies

Corresponding Author: Dr Jan Himmels

Version 0:

Reviewer comments:

Reviewer #1

(Remarks to the Author)

Review of "A systematic review of post COVID condition: Insights from Nordic population-based registry studies"

General:

- Major: search strategy is not available, and therefore could not be reviewed. While in the method section the authors state that they conducted a narrative review, the title is "systematic review". numbers in the PRISMA don't match the text:

Intro:

- First paragraph: consider adding references
- Line 35: how did you get this range from 3.1 to 20%? Is it for a specific geographic region? If so, please state it, if not consider adding a wider range, as there are higher and lower prevalence rates published worldwide
- Line 39: statements about bias need reference, or is it your personal opinion? Whole section needs re-writing. You could consider something like. In Europe/worldwide ranges are commonly reported between XY and XY. However, these reported ranges lack representativity/XY due to..
- Be consistent with the writing of COVID, line 52: covid, also line 427
- Line 51 to 55: repetition of "reduced by/due", consider rephrasing
- Line 65: " more chronic phase": kindly be more specific by adding a concrete timing
- Line 66ff consider taking out, redundant to line 70ff
- Line 74: do you really only mean long COVID here? If not add PCC

Methods:

- Line 419: was instead of is
- Line 422: what do you mean here?
- Where do you state that you only included Nordic registry studies – should be part of the inclusion criteria
- Line 429: doctor reported? What does this mean? E.g. lab confirmed? Clinical diagnosis? kindly be more specific in your topic of interest
- Kindly Include in your method section the definition for long COVID and PCC that you apply
- Line 447: if search strategy is previously used, please reference where
- Line 456: appendix not attached, search strategy was not reviewed.
- Line 459: please provide list of assessed websites as annex
- Line 472: two review authors, specify if they are the same as in line 470
- Line 478: self-explaining please take out
- Line 499: "we conducted a narrative review" – contradictory to title: systematic!!

Results:

- Line 81: citations? Don't you mean publications?
- Prisma: numbers in the PRISMA don't match the text: e.g. records screened 1827 vs. 1803
- If outcome e.g. line 122ff is not mentioned in table 1, please included reference of respective study in the text
- Line 153: "non-tested" as control group should be discussed
- Line 154: above you state there are studies that included participants from 2022, contradictory to statement Feb 2020- dec 2021. If you only mean those 8 studies on HC use, please be more specific

- Line 169: new comparator namely influenza patients? If you follow the methods outlined, this study should be excluded?
- Line 176: not clear to me "in covid-19 patients in negative controls"
- Kindly introduce abbreviation ICU
- Line 222: kindly delete s after infection
- Line 256: causation or correlation?
- Time frames are a little confusing in your result section, sometimes long term is already after 28 days, sometimes after more than 180, please add a respective transparent section in the method section
- Line 279: delete for

Discussion:

- Line 297, when does the subacute phase begin?
- Line 305: again, is it really a systematic review? You state narrative in the method section
- Kindly consider adding a section on associated factors in result and discussion
- Line 340, 392: abbreviation already introduced
- Line 343 and 345: reference missing
- It might be unclear to the reader, why testing over time becomes less accessible?
- Line 369: how is this statement supported by your results?

Conclusion:

Line 406: additional 2%? That would mean that 4% are still affected?

Reviewer #2

(Remarks to the Author)

Thank you for the opportunity to review this paper.

The authors set out to systematically review and synthesise Nordic registry studies which aimed to assess long-term service use and health outcomes of those with and without COVID-19. This paper is well written and clearly presented, however some additional detail is needed to allow for replication. Further discussion and consideration of the limitations of health record data is also needed.

Minor comments

1. Line 175. Please amend "was" to "were".
2. Line 354-359. The sentence starting "In later phases," is extremely long and difficult to follow. Please revise.

Specific comments:

Abstract

1. Line 8. The authors use the term "post-COVID condition (PCC)" in the abstract. Please define this in the introduction (e.g., a specific set of symptoms, following COVID-19 infection that last 12 or more weeks?)

Introduction

1. Line 30. Please provide a reference for the global prevalence rates of COVID-19 infection.
2. Line 36. An alternative reason for inconsistency between post covid condition (or long covid) prevalence rates stems from the differences between definitions. Please acknowledge this.
3. Line 53. What about those who don't present to health services? Are all individuals in Nordic countries registered with a health service? If not, you may still have a selected (and biased) sample. Further still, there may be those who are registered with a health service but do not present to health providers when experiencing ill health. Please add some detail (here or to the discussion) to reflect the limitations of registries.
4. Line 54. Can you please explain how confounding bias is reduced in registries through more thorough adjustment for socio-economic factors and calendar time? Do you mean as compared to cross-sectional studies? I don't believe that adjustment for IMD (for example) warrants a more thorough adjustment (as compared to the level of adjustment cohort studies). Please clarify how a more thorough adjustment is achieved using registry data.

Results and Discussion

1. Line 308. Here and in other places, the authors attribute the difference between PCC prevalence estimates (i.e., "3.1% up to 20%") to the mode of assessment (i.e., self-report vs registry data) and methodological differences between observational and registry data. However, there is a point missing around the drawbacks of reporting and coding PCC in primary care. For example, in the UK, we know that diagnostic codes for long COVID were instituted relatively late in the pandemic (2021-2022) and uptake by primary care practitioners was not uniform (see Walker et al. 2021; Thompson et al. 2022). Do you have any details about when the diagnostic codes for PCC were accessible in Nordic countries? Are these codes used uniformly across regions and countries? Please acknowledge limitation of relying on diagnostic codes when attempting to infer population prevalence.

Walker, A. J. et al. Clinical coding of long COVID in English primary care: a federated analysis of 58 million patient records in situ using OpenSAFELY. *Br. J. of Gen. Pract.* 71, e806–e814 (2021).

Thompson, E.J., Williams, D.M., Walker, A.J. et al. Long COVID burden and risk factors in 10 UK longitudinal studies and electronic health records. *Nat Commun* 13, 3528 (2022). <https://doi.org/10.1038/s41467-022-30836-0>

2. Line 311. Here the authors also specify that the "study covers entire populations". Is that true? As above, what about those

who don't present to health services? Please acknowledge that you may not be capturing specific sub communities or individuals (e.g., isolated communities, homeless or those with cultural preferences).

3. Line 311. Further, please clarify whether the studies you're reviewing are only capturing citizens of each country (i.e., seasonal workers, students, migrants and temporary residents, who may not also be registered with health services).

Methods

1. Line 433. More clarity about how the exposure group was defined is needed. How was exposure to COVID-19 ascertained? I.e., via a PCR test or antigen test or either?

2. Line 436. Also, more clarity about how the control group was defined is needed. Were controls defined by a negative test or a lack of wild type SARS-CoV-2?

3. 441. Please check if any of the preprints excluded have now been published and are eligible for inclusion.

4. Line 450. The authors searched for primary papers using Medline (part of PubMed database) and Embase (part of the Elsevier database), however they did not include any search engines which screen the Scopus or Web of Science database. I appreciate there is some overlap between databases, however WoS (for example) provides the largest coverage of journals (~40,000). Can the authors please add a rationale for only using these two databases and evidence that they have identified all relevant papers via these chosen databases. If unable to do so, can the authors please re-conduct the search within a larger set of databases to ensure they have not missed any key papers.

5. Line 456. I have not been able to view the appendix for this paper so cannot comment on the search terms. Please add these to a supplementary table.

Reviewer #3

(Remarks to the Author)

Review of the manuscript "A systematic review of post-COVID condition: Insights from Nordic population-based registry studies"

This manuscript is of great interest as it uses a narrative systematic review approach to combine and summarize findings across different papers on post-covid health problems.

Background The background provides a concise introduction to the theme, giving a rationale for using registry data.

Aim The aim is to compile and analyze data from Nordic registry studies to provide a comprehensive overview of the long-term effects of COVID-19. Further, this aim is broken down into perhaps six different sub-aims, which may not all be achieved?

"By synthesizing evidence from the Nordic countries, this review seeks to:

1. Identify common patterns and differences in long-term health outcomes.
2. Assess the burden of long COVID.
3. Highlight gaps in current knowledge.
4. Inform healthcare providers, researchers, and policymakers about the enduring impact of COVID-19.
5. Guide future research
6. -- and healthcare strategies to mitigate its long-term effects."

And the use of health care described at (line 148 ++) is not among the aims. And i.e the possibilities for future new strategies are poorly described. May be there should be somewhat better alignment between the sub aims and the results presented, and the conclusion?

Method

The method regarding search strategies and selection of papers is well described and is sound and in line with standards. Results They identified 22 papers from Nordic countries that were eligible according to the criteria they had set, all from the early period of the pandemic. The main content of the selected studies is detailed presented in the text.

However, it is unclear how PCC, used as the main finding, is "operationalized". In Table 1, the outcomes from the included papers are listed, and only 6 of the 22 studies have PCC as an outcome. How was this designed, and how does this relate to the definition of PCC without a common diagnosis code? The PCC probably relays on combining diagnoses, but what about using only symptoms? The PCC may be monosymptomatic, as an PCS? Summing up: how to differentiate PCC and PCS? I cannot find this distinction well described

Discussion The stipulated incidence was clearly in the lower part specter given in earlier studies with different methodology, and this is related to the use of registry data and not self-reporting.

However, (as also briefly mentioned) there are main limitations also with register data on health care as indicators of prevalences on population level. First, the patient must seek a doctor to be registered, and patients may not seek help, even if we can assume that those most bothered seek care. Another limitation is that doctors register one or more codes that they find of interest; symptoms may not be coded for patients with other diseases next, and we don't know how sensitive doctors are to report all symptoms. Additionally, there was no systematic registration of PCC as seen by doctors, since there was no specific code for this in use to be picked out of registries.

A strength is the use of registry data from Nordic countries that represents comparative settings supporting the ability to summarize findings.

It is interesting that even if the percentage is low regarding PCC the societal burden is high, this paper is not diminishing this health problems since numbers are high. This might be outlined even more clearly in the conclusion.

The paper is well structured, well written and using sound methods. The narrative approach used to extract findings from similar but still very different study-approaches performed in comparable countries and care systems is expanding the knowledge about health problems following covid infection. This paper clearly add useful knowledge to the still unclear and much debated field of prevalence of long time effects of a Covid-19 infection, both on individual and in societal level.

Reviewer comments:

Reviewer #1

(Remarks to the Author)

Thank you for the revised manuscript. The quality is now convincing.

Minor points:

- Intro:

- o Line 35: change to reporting a range between 10 and 20%.
- o Line 47: better post-acute COVID complaints?
- o Line 50 in brackets, consider trying to shorten to avoid interrupting the intro flow
- o Line 67: replace second "and" by "as well as"

- Results:

- o Line 112: consider adding a , e.g. 7,640 to improve readability
- o Fig 2: red mark ups should be deleted. Add software used to method section
- o Line 141: HC: is this abbreviation introduced?
- o Line 145 add , for better readability of big numbers
- o Line 188: replace 3 by the written word three
- o Line 198: add , for readability
- o Line 233: abbreviation already introduced
- o Line 242: the estimate *was

- Discussion:

- o Line 368: introduction of abbreviation needed

- Methods:

- o Line 485: did you develop the search strategy? Because in line 478 you state, that its previously used

Reviewer #3

(Remarks to the Author)

In the revised version of "A systematic review of post-COVID condition: Insights from Nordic population-based registry studies," the authors have precisely responded to my earlier remarks. The introduction of the new supplementary table 2 especially makes it easier for the reader to be oriented about the many variants of recording PCC and thereby interpret the results.

I have no further comments and will acknowledge the authors for their very well-structured and well-described revision.

We would like to thank the expert reviewers for their valuable input, which has helped to improve our manuscript. Please find below a point-to-point response to your comments and a list of the changes we made in the revised manuscript. The page and line references refer to the marked version of the manuscript.

Reviewer #1 (Remarks to the Author):

Comment	Our response	Action
General: - Major: search strategy is not available, and therefore could not be reviewed. While in the method section the authors state that they conducted a narrative review, the title is “systematic review”. numbers in the PRISMA don’t match the text:	Thank you for your detailed and constructive peer review. We are sorry it was not possible to access the search strategy which was uploaded in a research protocol to The Open Science Framework (OSF). We synthesised our data in a narrative manner but followed the strict systematic review methodology (including a priorly published protocol of our proposed approach). We mistakenly referred to our systematic review once as a narrative review, this shall be corrected. The PRISMA numbers were wrongly transferred from the screening software, the transparent software made this easy to correct, no studies were lost or overseen.	The search strategy is now attached as Appendix 1 and continues to be accessible from OSF (link), where the protocol is uploaded. We have also specified our methodological approach in the methodology section, under “Data management, analysis and synthesis”: “We conducted a systematic review, but as the heterogenous nature of identified study designs and reported outcomes did not allow for meta-analysis, data was synthesised narratively.” The PRISMA numbers were proofed and corrected.
Intro		
- First paragraph: consider adding references	We agree that further references may be useful.	Added reference to WHO “World Health Statistics 2024”, which shows that the COVID-19 pandemic reversed the trend of steady gain in life expectancy at birth and healthy life expectancy at birth (HALE).

- Line 35: how did you get this range from 3.1 to 20%? Is it for a specific geographic region? If so, please state it, if not consider adding a wider range, as there are higher and lower prevalence rates published worldwide	We agree that by not specifying region, it was unclear why that range was presented. We chose one example country, the UK and then for the European region. There are as you say wider ranges, but these are mainly reported in single studies and not as estimates for geographic regions.	We have specified the geographic regions in the text: “Estimates for prevalence of post covid condition (PCC) in the UK are around 3.1%, with WHO Europe reporting the range 10-20% (3,4)”
- Line 39: statements about bias need reference, or is it your personal opinion? Whole section needs re-writing. You could consider something like. In Europe/worldwide ranges are commonly reported between XY and XY. However, these reported ranges lack representativity/XY due to..	We think the background section needs to explicitly state which types of biases are commonly present in the existing literature on the post-covid condition, a reformulation addresses this. Our statements about bias are based on general and basic knowledge in epidemiological research, and we agree that this literature should be referred to.	We have added as a reference “Kenneth J. Rothman, Timothy L. Lash. Modern Epidemiology. 4th edition. 2015:1-768, i.e. this sentence now reads: “Inconsistencies in prevalence arise due to differences in case definitions and diagnostic strategies, but also as a result of study design. Observational studies that recruit participants based on their willingness to respond may be exposed to selection bias and recall bias when reporting typical post-covid symptoms (5-7).”
- Be consistent with the writing of COVID, line 52: covid, also line 427	We agree, thank you for pointing this out.	Changed covid to COVID in the whole text.
- Line 51 to 55: repetition of “reduced by/due”, consider rephrasing	We agree, thank you for pointing this out.	Replaced wording as suggested: “Misclassification bias is minimised by the greater reliance on objectively measured data, typically requiring clinical examination and doctor judgement for reporting post-COVID complaints.”
- Line 65: “more chronic phase”: kindly be more specific by adding a concrete timing	We agree that adding concrete timing reduces ambiguity.	Changed the phrasing: “However, existing data from these registries have never been combined to provide knowledge about the pandemic beyond 3 months.”
- Line 66ff consider taking out, redundant to line 70ff	We agree that there is redundancy, thanks for pointing that out.	Specified section taken out of the text.

- Line 74: do you really only mean long COVID here? If not add PCC	We agree that the formulation could be improved. Also, reviewer #3 suggested to revise this section, and we have decided to suggest a more substantial change. Please see comments and our response to reviewer #3	Reformulated the respective section as suggested by reviewer three, focusing on our primary outcomes. Please see action described for the first comment made by reviewer #3.
Methods		
- Line 419: was instead of is	Thank you for pointing out the misuse of tense.	Changed tense.
- Line 422: what do you mean here?	Line 422 reads “We did not exclude studies based on our assessment of methodological limitations. We did not use this information about methodological limitations to assess our confidence in the review findings.” The first sentence should be clear, but we understand that the second sentence can be more challenging. We meant to state that we did not take risk of bias into account in a system like GRADE, i.e. to rate our confidence in the results. However, we agree with the need for a more detailed description.	Revised to: “We did not exclude studies based on our assessment of methodological limitations, and we did not apply GRADE.”
- Where do you state that you only included Nordic registry studies – should be part of the inclusion criteria	We agree that it’s useful to specify this information also in the inclusion criteria.	Added “in the Nordic countries” at the end of the population description in inclusion criteria: “Persons tested for COVID-19 (not limited to organ-group, disease, or profession) in the Nordic countries”
- Line 429: doctor reported? What does this mean? E.g. lab confirmed? Clinical diagnosis? à kindly be more specific in your topic of interest	We agree that a single definition would be ideal, however given the heterogeneity of studies, we used the authors definitions. Narrowing this down may have led to identifying very few studies. We will include a new table with the definition used in each study.	We have included a new “Supplementary Table 2” under “Additional information” with the definitions and criteria used by the authors to define their populations and outcomes of interest.

- Kindly Include in your method section the definition for long COVID and PCC that you apply	Thank you for highlighting this. Reviewer #2 suggested to have it in the introduction. Because of similar reasons as described for the previous comment, we prefer to have the definition in the introduction section.	The WHO definition was added in the introduction. Please also see action to reviewer #2.
- Line 447: if search strategy is previously used, please reference where	We agree this is helpful and improves transparency.	Added relevant reference .
- Line 456: appendix not attached, search strategy was not reviewed.	We are sorry that the appendix was not in the shared file. The search strategy was initially only included in the openly accessible protocol. We agree that it should be added to the document for easy review.	Appendix 1 is now included at the end of the document.
- Line 459: please provide list of assessed websites as annex	Thank you for pointing this out, we agree.	The list of webpages we reviewed has now been added to Appendix 1 as part of the search strategy.
- Line 472: two review authors, specify if they are the same as in line 470	Thank you for pointing this out, yes they were the same.	Specified that they are the same authors.
- Line 478: self-explaining please take out	Thanks for noticing.	Removed the sentence.
- Line 499: “we conducted a narrative review” – contradictory to title: systematic!!	Thanks for pointing this out. We did a systematic review, but the results were synthesised narratively (because meta-analyses were not possible).	We changed the wording to reduce confusion about our methodological approach: “We conducted a systematic review, but as the heterogenous nature of identified study designs and reported outcomes did not allow for meta-analysis, data was synthesised narratively”
Results:		
- Line 81: citations? Don’t you mean publications?	Yes, thank you for pointing this out.	Corrected the term to “publications”.
- Prisma: numbers in the PRISMA don’t match the text: e.g. records screened 1827 vs. 1803	Thank you for noticing, we revisited EPPI reviewer where all screening decisions are tracked and found the mistake.	The numbers are corrected in the diagram, after cross-checking in our screening software. We also

		changed the word “reports” to “records” to use the same word throughout the diagram.
- If outcome e.g. line 122ff is not mentioned in table 1, please included reference of respective study in the text	We see that our formulation was misleading, we agree that consistent terminology use is important.	Removed “by reason or overall” for healthcare use, since it’s not part of the major characteristics listed in Table 1.
- Line 153: “non-tested” as control group à should be discussed	Thank you, we agree this was unclear.	We have revised this sentence to provide more clarity: “PCR positive cases were compared with negative controls (negative PCR in one control group and non-tested in another control group).” In addition, we have added the following to the discussion section, “Strengths and Limitations”: “In interpreting the findings, it is important to acknowledge that electronic health records in addition to implying type and severity of complaint or disease, also reflects healthcare-seeking behaviour. As such, it is possible that our findings apply mainly to individuals who seek care for their complaints. However, testing for the SARS-CoV-2 virus was free of charge and available to all citizens in the Nordic countries. Thus, healthcare seeking behaviour might also affect individuals’ PCR test patterns, where particularly health-conscious individuals might test themselves more often than non-health-conscious individuals. To take these differences in healthcare seeking behaviour and test patterns into account, some of the studies included in this review studied both a control group where individuals had a negative test and another control group where individuals had not tested themselves and were assigned a random test date. Generally, group differences in post-test healthcare use or diagnoses were greater in analyses including a non-

		tested control group (potentially overestimated) and smaller when including a group testing negative (potentially underestimated). The over- vs underestimation might be due to the control group’s differences in healthcare seeking behaviour relative to the group testing positive, which is likely somewhere in the middle with regard to healthcare seeking behaviour.”
- Line 154: above you state there are studies that included participants from 2022, contradictory to statement Feb 2020- dec 2021. If you only mean those 8 studies on HC use, please be more specific	Yes, we mean those 8 studies on HC use.	Specified in the text that the timeframe refers to the eight studies referred to earlier in the paragraph: “Individuals in these eight studies were included from February 2020 until December 2021”
- Line 169: new comparator namely influenza patients? If you follow the methods outlined, this study should be excluded?	We agree that this can appear as a new comparator, but since influenza patient are not COVID-19 positive they fall under relevant non COVID-19 controls. To reduce misinterpretation of the inclusion criteria, we will reformulate and better describe the applied PICO.	In the methodology section, under the PICO, we have provided a more detailed description of possible controls: “Non-COVID-19 controls: different definitions of non-COVID-19 controls were accepted as applicable to the outcome, including COVID-19 test-negative (with or without other infections) or untested” For further elaboration we have included a new “Supplementary Table 2” summarizing which definitions and criteria that were used to identify cases and controls in each of the included studies.
- Line 176: not clear to me “in covid-19 patients in negative controls”	We are sorry for these typos.	We have corrected to: “Skyrud et al. also reported that respiratory complaints were the main reason for the increased frequency of primary care visits that was seen in COVID-19 patients compared to negative controls (34).”
- Kindly introduce abbreviation ICU	Thanks for pointing this out.	Introduce abbreviation ICU.

- Line 222: kindly delete s after infection	Thanks for pointing this out.	Removed “s”.
- Line 256: causation or correlation?	We agree this was unclear.	We have changed the phrasing to “ association ”, to show that we are interested in any connection.
- Time frames are a little confusing in your result section, sometimes long term is already after 28 days, sometimes after more than 180, please add a respective transparent section in the method section	We agree this is confusing, primarily because the methods section does not explicitly state inclusion criteria regarding length of follow-up. We required at least 3 months of follow-up for inclusion in this review, however, this requirement did not prevent the included studies from following individuals already from day 0 after PCR testing or being ill with COVID-19.	We have now added to the inclusion criteria in the methods section, as part of the PICO: “Follow-up period: At least 3 months, however, this requirement did not prevent the included studies from following individuals already from day 0 after PCR testing or being ill with COVID-19.”
- Line 279: delete for	Thanks for noticing.	Deleted unnecessary word.
Discussion:		
- Line 297, when does the subacute phase begin?	We agree that being clearer on the timeframes makes it easier to follow.	In the methods section, under the PICO, we have added more information about follow-up period (see comment above), and also in the discussion section we now define the sub-acute stage as being 28days - 3 months post-infection whereas >3 months is defined as long term.
- Line 305: again, is it really a systematic review? You state narrative in the method section	Yes, it is a systematic review. A narrative data synthesis is used to synthesize result in systematic reviews when meta-analyses are inappropriate. On the other hand, a narrative review is often used to label non-systematic reviews. We understand that this can be confusing and have changed the wording.	We changed the wording to reduce confusion in the method section under: “Data management, analysis and synthesis”: “We conducted a systematic review, but as the heterogenous nature of identified study designs and reported outcomes did not allow for meta-analysis, data was synthesised narratively.”

- Kindly consider adding a section on associated factors in result and discussion	We agree, and we think that discussing the role of healthcare seeking behaviour in interpreting the results is important.	We have added a thorough discussion of how healthcare seeking behaviour might have influenced results, please see Discussion, “Strength and Limitations”: “In interpreting the findings, it is important to acknowledge that electronic health records in addition to implying type and severity of complaint or disease, also reflects healthcare-seeking behaviour. As such, it is possible that our findings apply mainly to individuals who seek care for their complaints. However, testing for the SARS-CoV-2 virus was free.....” Please also see our response to comment above regarding control groups (negative vs non-tested).
- Line 340, 392: abbreviation already introduced	Thanks for pointing it out.	Abbreviation removed.
- Line 343 and 345: reference missing	We are referring here to our own findings from all identified studies.	No reference added.
- It might be unclear to the reader, why testing over time becomes less accessible?	We agree that this may not be clear to all readers, we will change the text.	Specified that testing became less accessible, as it was neither recommended nor mandated, we also added a further reference (42) to substantiate the claim: “In later phases, testing became less accessible and is now neither recommended nor mandated. Simultaneously, vaccination rates have increased, and newer virus variants tended to cause milder disease (41, 42) with studies suggesting a lower risk of developing PCC (17).”
- Line 369: how is this statement supported by your results?	Thank you for highlighting this, we agree that this was a liberal interpretation of our findings. We will reformulate.	Deleted sentence and replaced it with: “Many of these patients needs professional care to get back to everyday life.”

Conclusion:		
Line 406: additional 2%? That would mean that 4% are still affected?	We agree that our formulation suggested your interpretation, but that was not well formulated, and we have no indication that it would be 4%.	We reformulated the section: "...., report an incidence of PCC in the general population to be less than 2%. This corresponds well with findings that most COVID-19 patients return to work within three months post-infection, with less than 2% of all COVID-19 patients still on sick leave after six months."

Reviewer #2 (Remarks to the Author):

Comment	Reply	Action
Thank you for the opportunity to review this paper. The authors set out to systematically review and synthesise Nordic registry studies which aimed to assess long-term service use and health outcomes of those with and without COVID-19. This paper is well written and clearly presented, however some additional detail is needed to allow for replication. Further discussion and consideration of the limitations of health record data is also needed.	We thank you for the critical and detailed review, the constructive feedback improves the quality overall. We agree our work could benefit from some more discussion regarding the limitations of health record data.	We have added a thorough discussion of how healthcare seeking behaviour might have influenced results, please see Discussion, “Strength and Limitations”: “In interpreting the findings, it is important to acknowledge that electronic health records in addition to implying type and severity of complaint or disease, also reflects healthcare-seeking behaviour. As such, it is possible that our findings apply mainly to individuals who seek care for their complaints. However, testing for the SARS-CoV-2 virus was free of charge and available to all citizens in the Nordic countries. Thus, healthcare seeking behaviour might also affect individuals’ PCR test patterns, where particularly health-conscious individuals might test themselves more often than non-health-conscious individuals. To take these differences in healthcare seeking behaviour and test patterns into account, some of the studies included in this review studied both a control group where individuals had a negative test and another control group where individuals had not tested themselves and were assigned a random test date. Generally, group differences in post-test healthcare use or diagnoses were greater in analyses including a non-tested control group (potentially

		overestimated) and smaller when including a group testing negative (potentially underestimated). The over- vs underestimation might be due to the control group’s differences in healthcare seeking behaviour relative to the group testing positive, which is likely somewhere in the middle with regard to healthcare seeking behaviour.” Please also comments made to Reviewer 1 under discussion above.
Minor comments:		
1. Line 175. Please amend “was” to “were”.	Thank you for pointing this out.	Changed to plural, from “was” to “were”.
2. Line 354-359. The sentence starting “In later phases,” is extremely long and difficult to follow. Please revise.	We agree that the sentence was a little too long, no meaning is lost by rewriting it.	Split the sentence in two and adjusted the formulation: “In later phases, testing became less accessible and is now neither recommended nor mandated. Simultaneously, vaccination rates have increased, and newer virus variants tended to cause milder disease (41, 42) with studies suggesting a lower risk of developing PCC (17).”
Abstract		
1. Line 8. The authors use the term “post-COVID condition (PCC)” in the abstract. Please define this in the introduction (e.g., a specific set of symptoms, following COVID-19 infection that last 12 or more weeks?)	We agree, it’s good to define, and do so early on.	Added definition and reference to the WHO definition : “It is defined as the continuation or development of new symptoms three months after the initial SARS-CoV-2 infection, with these symptoms lasting for at least two months with no other explanation (3).”

Introduction		
1. Line 30. Please provide a reference for the global prevalence rates of COVID-19 infection	We agree that further references may be useful.	Added reference to WHO Covid-19 dashboard, which provides an overview of COVID-19 incidence, associated deaths, etc.
2. Line 36. An alternative reason for inconsistency between post covid condition (or long covid) prevalence rates stems from the differences between definitions. Please acknowledge this.	We agree that acknowledging alternative reasons for inconsistencies strengthens transparency over where differences may come from.	Reformulated the sentence and added a further sentence: “Inconsistencies in prevalence arise due to differences in case definitions and diagnostic strategies, but also as a result of study design. Observational studies that recruit participants based on their willingness to respond are exposed to selection bias and recall bias when reporting typical post-covid symptoms (5-7).”
3. Line 53. What about those who don’t present to health services? Are all individuals in Nordic countries registered with a health service? If not, you may still have a selected (and biased) sample. Further still, there may be those who are registered with a health service but do not present to health providers when experiencing ill health. Please add some detail (here or to the discussion) to reflect the limitations of registries.	We agree that healthcare behaviour might be different for different individuals, i.e. that some persons would contact healthcare services with a certain symptom and others would not contact healthcare services with the same symptom.	We have added the following to the discussion section, please see under “Strengths and limitations”, or above in our response to your first comment.
4. Line 54. Can you please explain how confounding bias is reduced in registries through more thorough adjustment for socio-economic factors and calendar time? Do you mean as compared to cross-sectional studies? I don’t believe that adjustment for IMD (for	We agree this could have been better explained.	We have rephrased into: “Confounding bias can be reduced through thorough adjustment for objectively measured socioeconomic factors and calendar time (as many registry studies are designed based on a panel data setup, which enables the

example) warrants a more thorough adjustment (as compared to the level of adjustment cohort studies). Please clarify how a more thorough adjustment is achieved using registry data.		adjustment for calendar time in cases when time relative to the date of testing is studied).”
Results and Discussion		
1. Line 308. Here and in other places, the authors attribute the difference between PCC prevalence estimates (i.e., “3.1% up to 20%”) to the mode of assessment (i.e., self-report vs registry data) and methodological differences between observational and registry data. However, there is a point missing around the drawbacks of reporting and coding PCC in primary care. For example, in the UK, we know that diagnostic codes for long COVID we’re instituted relatively late in the pandemic (2021-2022) and uptake by primary care practitioners was not uniform (see Walker et al. 2021; Thompson et al. 2022). Do you have any details about when the diagnostic codes for PCC were accessible in Nordic countries? Are these codes used uniformly across regions and countries? Please acknowledge limitation of relying on diagnostic codes when attempting to infer population prevalence. Walker, A. J. et al. Clinical coding of long COVID in English primary care: a federated analysis of 58 million patient records in situ using	We agree this could have been better described and discussed. Diagnostic codes for PCC were accessible at different times in the Nordic countries. However, we believe it is also possible to infer the prevalence of PCC prevalence when studying group differences between individuals testing positive vs individuals testing negative or who are non-tested. If confounding is adequately adjusted for, a higher prevalence of diagnostic codes for fatigue among individuals who tested positive compared to those who tested negative 3–6 months post-testing can most likely be attributed to differences in SARS-CoV-2 infection. This group difference is independent of diagnostic coding practices because it relies on diagnostic codes that were also in use prior to the pandemic.	To the discussion section “Comparison to previous findings”, we have added: “In our study, differences in diagnostic coding practices of PCC across the Nordic countries might have affected the results. However, in studies comparing complaints as included in diagnostic codes already prior to the pandemic (e.g. ICPC-2 code A04 fatigue), group differences between individuals testing positive vs individuals testing negative or who are non-tested, also amounted to up to 2% (25). These data support the interpretation of a PCC prevalence of up to 2%, independent of diagnostic coding practices.”

OpenSAFELY. Br. J. of Gen. Pract. 71, e806–e814 (2021). Thompson, E.J., Williams, D.M., Walker, A.J. et al. Long COVID burden and risk factors in 10 UK longitudinal studies and electronic health records. Nat Commun 13, 3528 (2022). https://doi.org/10.1038/s41467-022-30836-0		
2. Line 311. Here the authors also specify that the “study covers entire populations”. Is that true? As above, what about those who don’t present to health services? Please acknowledge that you may not be capturing specific sub communities or individuals (e.g., isolated communities, homeless or those with cultural preferences).	We agree this should have been better discussed. The Nordic countries have a publicly funded healthcare system, which is available to everyone in the population, and when all citizens are included in the sample, the statement “..covering entire populations” is correct. Any underuse of healthcare services is likely not due to lack of availability – more likely, it is due to person characteristics, e.g. differences in healthcare seeking behaviour.	Because of the organization and structuring of the Nordic societies including their healthcare services, we would like to refrain from revising our statements regarding to whom our findings apply. However, we agree we could provide more discussion on how some people might be more prone than others to seek healthcare and how we believe this might have impacted on our findings. Please see our response and action to reviewer #2 comment#3 above, as well as discussion section, the second paragraph under “Strengths and Limitations”.
3. Line 311. Further, please clarify whether the studies you’re reviewing are only capturing citizens of each country (i.e., seasonal workers, students, migrants and temporary residents, who may not also be registered with health services).	We agree this could have been better described.	We have added to the discussion section, under “Comparison to previous studies”: “Where estimates of prevalence of PCC in previous studies are typically based on self-report, which induces bias due to self-selection and recall (5), the current study covers entire populations in entire geographical units with PCR testing available for everyone and routinely registered medical records for everyone. Some studies also included temporary workers or migrants, but there

		might be variations across studies, time periods and across the different Nordic countries in the extent to which these were included."
Methods:		
1. Line 433. More clarity about how the exposure group was defined is needed. How was exposure to COVID-19 ascertained? I.e., via a PCR test or antigen test or either?	We agree that a single definition would be ideal, however given the heterogeneity of studies, we used the authors definitions. Narrowing this down may have led to identifying very few studies. We will include a new table with the definition used in each study.	We have included a new “Supplementary Table 2” under additional information with the definitions and criteria used to identify cases and controls.
2. Line 436. Also, more clarity about how the control group was defined is needed. Were controls defined by a negative test or a lack of wild type SARS-CoV-2?	We agree that we could have been more transparent.	To make this clear we have included a table with definitions used by the authors. New “Supplementary Table 2” under can be found in the additional information, please see comment above.
3. 441. Please check if any of the preprints excluded have now been published and are eligible for inclusion.	The excluded preprint by O’Regan as listed in the PRISMA flow diagram is still not published as of 21.02.2025.	No change to text needed, as the study remains a pre-print.
4. Line 450. The authors searched for primary papers using Medline (part of PubMed database) and Embase (part of the Elsevier database), however they did not include any search engines which screen the Scopus or Web of Science database. I appreciate there is some overlap between databases, however WoS (for example) provides the largest coverage of journals (~40,000). Can the authors please add a rationale for only using these two databases and evidence that they have identified all	We agree that, in general, searching additional databases can help identify more relevant studies. However, we are confident that none have been missed, as our search strategy has been rigorously evaluated over several years in other similar research topics. In our latest report, we searched MEDLINE, EMBASE, Cochrane Central, Web of Science, the WHO COVID-19 Research Database, and Epistemonikos L·OVE. However, this did not provide an advantage over using	We have added the following as a strength to the discussion section, under “Strengths and Limitations” in the first paragraph: “Further, our search strategy, refined over several years and validated across multiple topics, minimizes the risk of missing relevant Scandinavian studies and strengthens the validity of findings.”

relevant papers via these chosen databases. If unable to do so, can the authors please re-conduct the search within a larger set of databases to ensure they have not missed any key papers.	only MEDLINE and EMBASE. Additionally, in previous reports, we applied this search strategy (with and without geographic limitations) and supplemented it with OpenAlex network searches covering 243 million works. These searches did not yield any additional relevant Scandinavian studies. Given the high level of connectivity within the Scandinavian research community—for example, through annual Nordic researcher gatherings organized by NordForsk—we have a comprehensive overview of relevant publications. Furthermore, we proactively reached out to researchers in the field to ensure no relevant studies were overlooked. We hope this reassures you that our approach has minimized the risk of missing relevant studies.	
5. Line 456. I have not been able to view the appendix for this paper so cannot comment on the search terms. Please add these to a supplementary table.	We are sorry that the appendix was not in the shared file, the strategy is included in the openly accessible protocol, but we agree that it should be added to the document for easy review.	Appendix 1 is now included at the end of the document.

Reviewer #3 (Remarks to the Author):

This manuscript is of great interest as it uses a narrative systematic review approach to combine and summarize findings across different papers on post-covid health problems. Background The background provides a concise introduction to the theme, giving a rationale for using registry data.

Comment	Reply	Action
Aim The aim is to compile and analyze data from Nordic registry studies to provide a comprehensive overview of the long-term effects of COVID-19. Further, this aim is broken down into perhaps six different sub-aims, which may not all be achieved? “By synthesizing evidence from the Nordic countries, this review seeks to:  1. Identify common patterns and differences in long-term health outcomes. 2. Assess the burden of long COVID. 3. Highlight gaps in current knowledge. 4. Inform healthcare providers, researchers, and policymakers about the enduring impact of COVID-19. 5. Guide future research 6. -- and healthcare strategies to mitigate its long-term effects.” And the use of health care described at (line 148 ++) is not among the aims. And i.e the possibilities for future new strategies are poorly described. May be there should be somewhat better alignment between the sub aims and the results presented, and the conclusion? 	We agree that the aim text clearly benefits from some revision and streamlining with primary outcomes.	We reduced the bloaty description of aims to the primary outcomes: “By synthesizing evidence from the Nordic countries this review seeks to identify the incidence of PCC, the associated healthcare use and sick leave, and new onset diseases.”

Method		
The method regarding search strategies and selection of papers is well described and is sound and in line with standards.	Thank you.	
Results		
They identified 22 papers from Nordic countries that were eligible according to the criteria they had set, all from the early period of the pandemic. The main content of the selected studies is detailed presented in the text.	Thank you.	
However, it is unclear how PCC, used as the main finding, is “operationalized”. In Table 1, the outcomes from the included papers are listed, and only 6 of the 22 studies have PCC as an outcome. How was this designed, and how does this relate to the definition of PCC without a common diagnosis code? The PCC probably relays on combining diagnoses, but what about using only symptoms? The PCC may be monosymptomatic, as an PCS? Summing up: how to differentiate PCC and PCS? I cannot find this distinction well described	We agree with you, as well as both other reviewers that it is important to be more transparent about how definitions were used and operationalised. We believe that this is best addressed through a new table which specifies for each study how the authors thereof chose to define or identified their population of interest.	Added a new table “Supplementary Table 2” under “Additional information”, which provides an overview of how definitions were used and operationalised. We refer to “Supplementary Table 2” in the text and in Table 1. Further, we have included some more discussion on the PCC vs PCS in the included studies, see under “Comparison to previous studies”: “In our study, differences in diagnostic coding practices of PCC across the Nordic countries might have affected the results. However, studies comparing complaints included in diagnostic codes available prior to the pandemic (e.g. ICPC-2 code A04 fatigue), also found a 2% increase in individuals testing positive for SARS-CoV-2 (25). These data

		support the interpretation of a PCC prevalence of up to 2%, independent of diagnostic coding practices.
Discussion		
The stipulated incidence was clearly in the lower part specter given in earlier studies with different methodology, and this is related to the use of registry data and not self-reporting. However, (as also briefly mentioned) there are main limitations also with register data om health care us as indicators of prevalences on population level. First, the patient must seek a doctor to be registered, and patients may not seek help, even if we can assume that those most bothered seek care. Another limitation is that doctors register one or more codes that they find of interest; symptoms may not be coded for patients with other diseases next, and we don't know how sensitive doctors are to report all symptoms. Additionally, there was no systematic registration of PCC as seen by doctors, snice there was no specific code for this in use to be picked out of registries.	We agree with the reviewer. Please, also see responses to reviewer 2, comment #1, #2, and #3 under his/hers heading “Results and discussion”.	Please see our discussion section and response to the previous comment. We have also added a more thorough discussion regarding healthcare-seeking behaviour, under “Strength and Limitation”: “In interpreting the findings, it is important to acknowledge that electronic health records in addition to implying type and severity of complaint or disease, also reflects healthcare-seeking behaviour. As such, it is possible that our findings apply mainly to individuals who seek care for their complaints. However, testing for the SARS-CoV-2 virus was free of charge and available to all citizens in the Nordic countries. Thus, healthcare seeking behaviour might also affect individuals’ PCR test patterns, where particularly health-conscious individuals might test themselves more often than non-health-conscious individuals. To take these differences in healthcare seeking behaviour and test patterns into account, some of the studies included in this review studied both a control group where individuals had a negative test and another control group where individuals had not tested themselves and were assigned a random test date. Generally, group differences in post-test healthcare use or

		diagnoses were greater in analyses including a non-tested control group (potentially overestimated) and smaller when including a group testing negative (potentially underestimated). The over- vs underestimation might be due to the control group’s differences in healthcare seeking behaviour relative to the group testing positive, which is likely somewhere in the middle with regard to healthcare seeking behaviour.”
A strength is the use of registry data from Nordic countries that represents comparative settings supporting the ability to summarize findings.	Thank you. We think this strength should be added to our discussion section.	Now added: “An important strength of our work is the use of registry data from Nordic countries that represents comparative settings supporting the ability to summarize findings.”
It is interesting that even if the percentage is low regarding PCC the societal burden is high, this paper is not diminishing this health problems since numbers are high. This might be outlined even more clearly in the conclusion.	We agree, a small percentage of society implies that many people are suffering. The change over time makes it difficult to estimate how many people are affected as of now. We believe that by stating “our results indicate that a significant number of patients continues to experience long-lasting societal burden beyond three months” we acknowledge the burden sufficiently.	Reformulated the conclusion to: “Nordic registry studies, characterized by limited selection bias and minimal time-varying confounding, report an incidence of PCC in the general population to be less than 2%. This corresponds well with findings that most COVID-19 patients return to work within three months post-infection, with less than 2% of all COVID-19 patients still on sick leave after six months. While typical post-COVID complaints, healthcare utilization and sick leave decline rapidly during the sub-acute phase, our results indicate that a significant number of patients continues to experience

		long-lasting societal burden beyond three months. Our findings may imply that health- and welfare services may need to be up-scaled to address this challenge. Furthermore, improved and earlier clinical management of PCC could help reduce the future strain on healthcare systems and the workforce."
The paper is well structured, well written and using sound methods. The narrative approach used to extract findings from similar but still very different study-approaches performed in comparable countries and care systems is expanding the knowledge about health problems following covid infection. This paper clearly add useful knowledge to the still unclear and much debated field of prevalence of long time effects of a Covid-19 infection, both on individual and in societal level.	Thank you!	

We would like to thank the expert reviewers for their valuable input, which has helped to improve our manuscript. Please find below a point-to-point response to your comments and a list of the changes we made in the revised manuscript. The page and line references refer to the marked version of the manuscript.

Reviewer #1 (Remarks to the Author):

Comment	Our response	Action
Thank you for the revised manuscript. The quality is now convincing. Minor points:	Thank you for your additional corrections and suggestions. It's greatly appreciated to receive such rigorous and concrete feedback, which helps ensure both high quality and a clear, engaging reading experience.	
Intro:		
o Line 35: change to reporting a range between 10 and 20%.	Thanks for pointing this out.	Changed.
o Line 47: better post-acute COVID complaints?	Yes, that formulation is more precise, we will change it in the text.	Changed.
o Line 50 in brackets, consider trying to shorten to avoid interrupting the intro flow	Good point, it wasn't good style to have a bracket longer than the main sentence. Rephrased, and removed the brackets.	Reformulated the sentence, leaving the brackets away, whilst reducing the length overall: “Confounding bias can be reduced through thorough adjustment for objectively measured socioeconomic factors and calendar time, which is often addressed through panel data setups in registry studies.”

o Line 67: replace second “and” by “as well as”	Good point, we agree to replace the word.	Replaced.
Results:		
o Line 112: consider adding a , e.g. 7,640 to improve readability	Well spotted, large numbers should be written with commas to separate thousands, this was not consistent throughout the document. Commas will be added where necessary.	Large numbers missing commas to separate thousands were inserted and marked in the text.
o Fig 2: red mark ups should be deleted. Add software used to method section	Well spotted, the red marks are inappropriate and should be removed. The figure is a table, with overlaying shapes made within MS Office/Word.	Removed red markings. Added to the methodology section: “ All figures and tables were created with MS Office ”.
o Line 141: HC: is this abbreviation introduced?	It was not, thank you. We think its best to remove the abbreviation from the title and move it into the text.	Removed abbreviation from title. Moved abbreviation to the text. Highlighted in the document.
o Line 145 add , for better readability of big numbers	See also comment above, all missing commas for large numbers were added.	Large numbers missing commas to separate thousands were inserted and marked in the text.
o Line 188: replace 3 by the written word three	Thank you for noticing, change number to written out form	Changed to written word.
o Line 198: add , for readability	See also comment above, all missing commas for large numbers were added.	Large numbers missing commas to separate thousands were inserted and marked in the text.
o Line 233: abbreviation already introduced	Thanks!	Removed already introduced abbreviation as suggested.
o Line 242: the estimate *was	Thanks!	Changed to correct tense.
Discussion:		

Line 368: introduction of abbreviation needed	Well spotted. Agreed, its only mentioned once in the text, so its best to write it out this time.	Written out in the text as: "general practitioner"
Methods:		
Line 485: did you develop the search strategy? Because in line 478 you state, that its previously used	Yes, we developed the search strategy previously and then adopted it for this article. We recognise that the formulation in line 485 could be misunderstood, as the whole sentence does not provide new information, it can be deleted.	Deleted the sentence: " We developed search strategies for each database. "

Reviewer #3 (Remarks to the Author):

Comment	Reply	Action
In the revised version of "A systematic review of post-COVID condition: Insights from Nordic population-based registry studies," the authors have precisely responded to my earlier remarks. The introduction of the new supplementary table 2 especially makes it easier for the reader to be oriented about the many variants of recording PCC and thereby interpret the results. I have no further comments and will acknowledge the authors for their very well-structured and well-described revision.	We thank the reviewer for the positive feedback and for acknowledging the revisions made to the manuscript. We are pleased that the addition of Supplementary Table 2 helped clarify the different approaches to recording post-COVID condition and improved the interpretability of our findings. We appreciate the reviewer's thoughtful comments throughout the review process.